# Life Expectancy and Mortality in the Aspect of Diverse Environmental Exposure to PCDD/Fs and PCBs—Ecological Case Study from the Silesia Province, Poland

**DOI:** 10.3390/toxics13111002

**Published:** 2025-11-20

**Authors:** Aleksandra Duda, Agata Piekut, Grzegorz Dziubanek

**Affiliations:** 1Doctoral School, Medical University of Silesia in Katowice (Poland), 15 Poniatowskiego Street, 40-055 Katowice, Poland; d200975@365.sum.edu.pl; 2Department of Environmental Health, School of Public Health in Bytom, Medical University of Silesia in Katowice (Poland), 18 Piekarska Street, 42-902 Bytom, Poland; 3Department of Environmental Health Risk Factors, School of Public Health in Bytom, Medical University of Silesia in Katowice (Poland), 18 Piekarska Street, 42-902 Bytom, Poland; gdziubanek@sum.edu.pl

**Keywords:** MLE, MR, inhalation exposure, PCDD/Fs, PCBs, Silesia Province, Poland, environmental epidemiology, public health risk assessment

## Abstract

The present ecological study endeavours to evaluate the variability of life expectancy (MLE) and mortality rate (MR) on a micro scale, specifically between populations of neighbouring villages in the Silesia Province. This endeavour is of particular significance due to the chronic exposure to halogenated persistent organic pollutants (POPs) in one of the villages under study. The present study is innovative in comparison with previous research in this field, as it considers the impact of the most toxic dioxin, 2,3,7,8-tetrachlorodibenzo-p-dioxin (2,3,7,8-TCDD), and utilises a village-level reference area. A thorough investigation was undertaken to determine the possible consequences of inhalation exposure within the local community to polychlorinated dioxins and furans (PCDD/Fs) and polychlorinated biphenyls (PCBs). A robust correlation was evidenced between chronic exposure of case study residents to 2,3,7,8-TCDD and their mortality. Furthermore, an investigation revealed a strong correlation between the concentration of 2,3,7,8-TCDD in the air and actual MLE. An increase in the concentration of 2,3,7,8-TCDD by 10 fg I-TEQ/m^3^ has the potential to result in a reduction in the mean MLE of the exposed inhabitants of Silesia by 1 year and 9 months. In addition, the results of this study indicate that the female population exhibits a lifespan that is 7 years and 10 months longer than that of the male population. However, given the substantial differences in the mean MLE observed also in low-polluted areas of this region, it is probable that not only environmental factors, including exposure to PCDD/Fs and PCBs, but also various socio-economic factors may be involved.

## 1. Introduction

Air pollution has been demonstrated to be a contributing factor to a wide range of health complications on a global scale, irrespective of socio-economic status. Numerous studies have documented the relationship between long-term exposure to air pollutants and the adverse health effects that ensue, thus leaving no doubt as to the scale of the health risk posed by the airborne chemicals [1,2]. Research has demonstrated that exposure to atmospheric particulate matter pollution exerts an influence on mortality from all causes, as well as from cardiovascular and respiratory causes [3,4]. According to estimates by the World Health Organisation (WHO), air pollution was responsible for 4.2 million premature deaths worldwide in 2019. The mortality rate (MR) was found to be directly attributable to exposure to fine particulate matter (PM), which has been linked to the development of cardiovascular and respiratory diseases, as well as cancer [5]. In Poland (PL), a considerable proportion of the country’s territory experiences air quality that is among the poorest in the European Union (EU). As indicated in the European Environment Agency (EEA) report, in the period 2021–2023, 23 of the 50 most polluted cities in the EU were located on Polish territory, with seven of these cities ranking in the top ten. Consequently, Poland experiences an annual excess of 45,000 premature mortalities, attributable in indirect measure to low emissions [6,7]. Low-emission sources are defined as those that emit a maximum of 30 m above ground. These sources are typically associated with the municipal and residential sectors. The findings of the World Air Quality Report 2020 unequivocally designate Orzesze (a municipality within the Silesia Province) as the most contaminated city in Europe, with an annual mean PM2.5 concentration of 44.1 µg/m^3^ [8]. This is undoubtedly due to one of the worst energy source structures in the EU. In 2021, the share of solid fossil fuels in Poland’s energy mix was 43%, whereas the EU’s share was 12.4% [9]. Silesia Province has been identified as the most industrialised administrative area in Poland. The region has a long history of coal and non-ferrous metal mining, which has contributed to its economic development and industrialisation. Despite the high concentration of the industry in Silesia, the primary concern at present is the emission of particulate pollutants into the atmosphere from residential dwellings heated with coal, the most economical and readily available energy source in the region [10,11]. The issue is of particular significance in locations where, with a view to enhancing the energy efficiency of lower-grade coal (referred to as ‘culm’ or ‘coal dust’), plastic waste is frequently incinerated alongside the coal. This has been shown to result in increased emissions of toxic compounds into the atmosphere, including particulate matter (PM), sulphur dioxide (SO_2_), dioxins (PCDDs), furans (PCDFs), and polychlorinated biphenyls (PCBs) [12,13]. In the realm of PCDD compounds, the congener 2,3,7,8-tetrachlorodibenzo-p-dioxin (2,3,7.8-TCDD) merits particular consideration due to its unequivocal and substantiated toxicological impact on exposed populations, as evidenced by a plethora of studies. It is indicated by a substantial body of research that PCDDs/PCDFs and PCBs have the potential to compromise human health even at low exposure levels. It has been demonstrated that chronic exposure has been associated with a range of adverse effects, including, but not limited to, fertility and reproductive disorders, congenital anomalies, fetal death, and endocrine disruption. Furthermore, exposure to these compounds has been demonstrated to result in immune system disorders and the development of malignant tumours. In view of these concerns, the absence of air concentration monitoring for these pollutants is problematic [3,5,7,14].

The issue in Poland is of particular significance due to the absence of continuous monitoring of the emissions of the most toxic compounds released into the atmosphere, PCDD/Fs and PCBs. These substances are released in substantial quantities during the process of co-incineration. As demonstrated in previous studies, the concentration of these compounds in the cities of the Silesia region ranged from 78.3 to 320.5 fg I-TEQ/m^3^ [13].

As part of this research, the case study area selected was the municipality of Koziegłowy (KZ), with a particular focus on the village of Cynków (CY) in the Silesia Province. In this area, we could indicate chronic inhalation exposure to high concentrations of toxic air pollutants emitted from low-emission sources. This is attributable to the fact that for a period exceeding three decades, local residents have been engaged in the production of polyvinyl chloride (PVC) products (e.g., Christmas trees, wreaths, flowers, etc.) and utilising the production waste as a fuel source in domestic heating systems and boiler houses. This has resulted in the creation of conditions conducive to the release of PCDD/Fs and PCBs. The capacity to analyse data pertinent to this population constitutes a distinctive and pioneering aspect of the present study. The case study area is situated between the Upper Silesia Agglomeration (an agglomeration of European importance) and the city of Czestochowa (see map in Figure 1). The area is located within a temperate climate zone, characterised by an average temperature of 9.0 °C, precipitation of 769 mm/year, and a predominant wind direction of south-west [15]. In the designated study area, the presence of heavy industry, waste incineration facilities, and other industrial sources emitting polyhalogenated persistent organic compounds, like PCDD/Fs and PCBs, has been ruled out [13]. The present study constitutes an endeavour to augment the extant corpus of knowledge on the subject by means of a comparison of MR and MLE in a selection of villages within the municipal boundaries of Koziegłowy (KZ).

MR and MLE are key indicators of population health and are an essential component of public health strategies aimed at continuously improving societal health. Analysing negative health metrics enables us to recognise the extent of health inequalities in the population and assess its health status [16]. Death provides key information on health deficiencies in the population, and mortality statistics are the primary source of information on the distribution of life risks caused by serious diseases or external factors. In 2019, the WHO recorded an average life expectancy (MLE) of 73.4 years for the global population, an increase of more than six years since 2000 (66.8 years) [5]. MLE in the EU increased steadily until 2019, reaching 81.3 years. However, a subsequent decrease was observed, with MLE falling to 80.1 years in 2021. The average MLE for EU women was 82.9 years, which was 5.7 years longer than for men at 77.2 years [17]. The average MLE of the Polish population is significantly lower than the European average. According to the latest data from the Central Statistical Office (CSO) in 2021, the average female MLE in Poland was 79.7 years. The highest MLE was recorded in the Lesser Poland Province at 80.9 years, while one of the lowest was recorded in the Silesia Province at 78.9 years. The male population of Poland recorded an average MLE of 71.7 years, with the highest MLE observed in the Lesser Poland Province at 70.1 years. In 2021, the average male MLE in Silesia was 71.2 years [17,18]. In 2021, the MR in the EU was 11.8 per 1000 people, which was 0.2 and 1.2 higher than in 2020 and 2019, respectively [19]. Meanwhile, in Poland, it was 13.6 per 1000 people [20]. In the Silesia Province, however, mortality stood at 15 per 1000 inhabitants in 2021 [21].

One of the most important reasons for existing inequalities in MLE and MR in PL compared to the EU may be air pollution. Therefore, it is extremely valuable to assess the variation in these parameters at a micro level, i.e., between populations of villages close to each other. This is particularly important given that the case study village has experienced significant and long-lasting exposure to selected persistent organic pollutants (POPs). The results of this study could inform the planning of preventive measures, such as the implementation of advanced methods, techniques, solutions, and materials for air pollution control [22,23]. In the selected villages, the potential impact of the magnitude of the local community’s inhalation exposure to selected halogenated POPs on the MR and MLE of the population was analysed. This study aims to analyse the magnitude of exposure to PCDD/Fs and PCBs in the populations of these areas, and to determine the impact of the analysed risk factors on MLE and MR in these populations.

## 2. Materials and Methods

### 2.1. Study Area

The study selected two neighbouring villages, Cynków (CY) and Gniazdów (GN), which are located in the municipality of Koziegłowy (KZ) in the southern part of Poland (50°35′59.89″ N, 19°09′46.50″ E). The residents of these villages have been exposed to toxic compounds such as PCDD/Fs and PCBs for several decades due to the uncontrolled burning of a mixture of post-production plastic waste and low-quality fossil fuels. Between 2005 and 2023, CY municipality had an average population of 1175, while GN municipality had an average population of 915, which is less than 15% of the 14,113 people in KZ municipality. In the case study area, women represented 51.8% of the population, while men represented 48.2%. To enable comparison, a reference population of inhabitants from Kochcice (KO) village was determined. This village had an average population of 1880 between 2005 and 2023. It is situated in the Kochanowice (KC) municipality, which has an average population of 6862 and is located approximately 50 km from the study population. Apart from the low-emission sources typically found in Poland (Figure 1), this municipality has no other sources of exposure to PCDD/Fs and PCBs. In the control population, females accounted for 51.2% of the population, while males accounted for 48.8%. Using a reference area at the village level considerably increases the precision of the present study compared with the previous study conducted in this location. To evaluate the impact of chronic inhalation exposure to PCDD/Fs and PCBs on the mean real MLE and MR (excluding deaths from external causes) of residents in designated villages and municipalities, data from 19 cities with county rights in Silesia Province were included in the calculations. The methodology for calculating MLE and MR is explained in the following paragraph.

### 2.2. Data Sources

Data on the population and number of deaths by gender from 2005 to 2022 were obtained from the Central Statistical Office (CSO) in Poland, as well as from the databases of the local statistical offices in the surveyed villages and communes. The acquired death data excludes all deaths due to external causes, as defined by the International Classification of Diseases (ICD-10). According to the ICD-10, these include accidents, suicides, assaults, and events of undetermined intent [24]. The obtained data relates to:the population of three villages (CY, GN and KO) and two municipalities (KZ and KC), which constitute the case study area.19 cities with county rights in the Silesian Voivodeship: Bielsko-Biała (BB), Bytom (BY), Chorzów (CH), Częstochowa (CZ), Dąbrowa Górnicza (DG), Gliwice (GL), Jastrzębie-Zdrój (JZ), Jaworzno (JA), Katowice (KA), Mysłowice (MY), Piekary Śląskie (PS), Ruda Śląska (RS), Rybnik (RY), Siemianowice Śląskie (SS), Sosnowiec (SO), Świętochłowice (SW), Tychy (TY), Zabrze (ZA) and Żory (ZO).

The mean MLE was calculated for all study populations between 2005 and 2022, disaggregated by gender, using the formula obtained from the data:$$\mathrm{MLE}\;=\;\frac{\begin{array}{c} sum\;of\;years\;lived\;by\;women\;and\;men\;who\;died\;in\;the\;year\;\\ (excluding\;deaths\;from\;external\;causes)\; \end{array}}{number\;of\;women/men\;who\;died\;in\;the\;year\;under\;review\;}$$

Furthermore, MR was calculated for the populations of socio-districts (CY, GN and KO) and municipalities (KZ and KC) between 2005 and 2022, broken down by gender, using the following formula:$$\mathrm{MR}\;=\;\frac{\begin{array}{c} number\;of\;deaths\;of\;women/men\;in\;a\;given\;year\;\\ \left(excluding\;external\;causes\right) \end{array}}{population\;of\;woman/men\;in\;the\;same\;year\;}\times 1000$$

### 2.3. Air Sampling

To determine the extent of exposure to PCDD/Fs and PCBs among the populations under study, air sampling was conducted over two periods:February–March 2021: villages CY, GN and KO; cities BB, BY, CH, DG, GL, KA, PS, RS, RY, SS, SO and ZA.December 2022–January 2023: villages CY, GN and KO; municipalities KZ and KC.

Passive air sampling was conducted using 14 cm diameter, 1.5 cm thick PUF (polyurethane foam) sorbents (E&H Services, a.s., Frýdek-Místek, Czech Republic), as described in Kohoutek et al. [25]. The samplers were suspended in free-flowing areas approximately 2 m above the ground for 36 days. The sampling volume was calculated using a conversion factor based on an assumed daily airflow of 3.5 m^3^/day [26,27]. Prior to air sampling, the PUF discs were cleaned using Soxhlet extraction. Toluene was used as a solvent for the first 24 h, followed by acetone for the next 24 h and then toluene for the final 8 h. For this study, we also used the results of PCDD/F and PCB concentration measurements in air samples from previous years:from December 2014 to February 2015, the following areas were covered: the village of CY, the municipality of KZ, and the cities of BB, BY, CH, CZ, DG, GL, JA, KA, MY, PS, RS, RY, SO, SW, TY and ZA.from November 2015 to February 2016: the villages of CY and the towns of BY, CH, DG, GL, KA, TY and ZA.

The effect of the population’s inhalation exposure to persistent organic compounds on real MLE was calculated using the results.

### 2.4. Air Sample Analysis

During production, the XAD 2 cartridges (a hydrophobic copolymer of styrene-divinylbenzene resin) and the PUF cartridges (polyurethane foam) were spiked with a sampling standard (EN-1948SS, Wellington Laboratories, Guelph, ON, Canada) [28], which determines whether the sampling was carried out correctly. In the laboratory, PCDD/F and PCB concentrations were determined by performing chemical analyses using isotope dilution gas chromatography–mass spectrometry based on analytical procedure P01/03 in an accredited testing laboratory (E&H Services, a.s., Frýdek-Místek, Czech Republic) in accordance with CSN EN ISO/IEC 17025:2018 [29]. The samples were spiked with 13C12-labelled internal standards (EN-1948ES, MBP-CP, MBP-MO and MBP-MXE, Wellington Laboratories, Canada) [30]. The PUF filters and XAD 2 were then extracted together in a Soxhlet apparatus for 24 h in toluene. Once extraction was complete, the extract was evaporated to a volume of approximately 3–5 mL using an evaporator (Turbovap II, Zymark Corp., Hopkington, MA, USA). The samples were then transferred quantitatively to a separating funnel and purified with sulphuric acid until the solution was clear. The purified samples were then washed with distilled water and filtered through a layer of sodium sulphate. The samples were concentrated to a volume of about 1 mL using RVO (Turbo Vap, Caliper Life Sciences, Hopkinton, MA, USA). Next, the silica gel (basic and acidic) was cleansed in the chromatographic column and the extract was concentrated using RVO (Turbo Vap, Caliper LifeSciences, Hopkinton, MA, USA) to a volume of about 1 mL. The sample was then purified on alumina and activated carbon. PCDD/Fs, dl-PCBs (dioxin-like polychlorinated biphenyls) and ndl-PCBs (non-dioxin-like polychlorinated biphenyls) were determined using gas chromatography with triple quadrupole GC-MS/MS (GC—Trace 1300, MS—TSQ 9000, Thermo Fisher Scientific, Waltham, MA, USA), using the following columns: Rxi-5Sil-MS (60 m × 0.25 mm × 0.25 µm; helium as the carrier gas). At the end of the analytical procedure, a syringe sample was added to correct the gas chromatography response (EN-1948IS and MBP-70, Wellington Laboratories, Guelph, ON, Canada) [31].

Quantification was performed using the internal standard method. This method is based on the comparison of the chromatographic peak areas of the labelled internal standards with those of the analytes. The recovery of the internal standards (PCDD/Fs, dl-PCBs and ndl-PCBs) was in the range 25–130%. The concentrations of substituted congeners were quantified based on the isotope ratio to within ±30% of the theoretical values, with a signal-to-noise ratio of ≥3.

These results were then used to calculate the average daily exposure to 2,3,7,8-tetrachlorodibenzo-p-dioxin (2,3,7,8-TCDD), PCDD/Fs, dl-PCBs and n-dl-PCBs. The air content of PCDD/Fs, 2,3,7,8-TCDD, dl-PCBs and n-dl-PCBs was determined using the formula outlined by Dziubanek et al. [13]:


C_Air_ = C_sample_/(R_s_ × Ds.)


C_Air_—air content (fg I—TEQ m^3^/day)C_sample_—sample content (fg I—TEQ/sample)R_s_—sampling rate (m^3^/day)Ds.—sampling duration (day)

### 2.5. Statistical Analysis

The study involved conducting a statistical analysis of the results in several stages. Firstly, Pearson’s correlation coefficients were calculated for the time of the study (in years) and the MR and MLE. These were then statistically verified using the *t*-test. Secondly, the study examined whether there were significant differences in the MR and MLE between the villages and municipalities studied. The Shapiro–Wilk test was used to verify the applicability of parametric tests in this analysis by testing the normality of the distribution. Non-parametric tests were then used to confirm the absence of normality: the Mann–Whitney test for comparisons between municipalities, and the Kruskal–Wallis test for comparisons between villages. The normality of the distributions was positively verified by applying the *t*-test for two independent samples to compare municipalities and the one-way ANOVA test to compare villages. Before conducting the one-way ANOVA, we tested for homogeneity of variance using Levene’s test. As homogeneity of variance was not confirmed, the Brown–Forsythe F* test was used instead. Post hoc tests were used where statistical significance was found:Dun–Bonferroni and Conover tests (test Kruskal–Wallis),LSD Fisher and Sheffe tests (One-way Anova),Brown–Forsythe and Games–Howell tests (Brown–Forsythe F* test)

In the third step, calculations were performed to determine multiple regression equations for MR and MLE in relation to PCDD/Fs, dl-PCBs and ndl-PCBs concentrations. The significance of the regression coefficients was verified using the *t*-test, and the coefficient of determination (R^2^) was verified using the F-test.

## 3. Results

### 3.1. The Concentration of Polychlorinated Dibenzo-p-Dioxins and Dibenzofurans (PCDD/Fs) and Polychlorinated Biphenyls (PCBs) in the Case Study Area and Silesia Province

In a study examining the population’s exposure to PCDD/Fs in municipalities with district rights in Silesia in 2021, ZA had the highest average daily concentration of 484.1 fg I-TEQ/m^3^, while SO had the lowest of 182.5 fg I-TEQ/m^3^. Similarly, the highest concentration of the most toxic dioxin compound, 2,3,7,8-TCDD, was recorded in ZA at 134.9 fg I-TEQ/m^3^, and the lowest concentration was found in SO at 41.3 fg I-TEQ/m^3^. Exposure to dl-PCBs in the population of the considered cities was highest in ZA at 166.7 fg I-TEQ/m^3^, while the lowest exposure was observed in two cities, BB and BY, at 30.2 fg I-TEQ/m^3^. Daily mean ndl-PCB concentrations ranged from a maximum of 6.7 pg/m^3^ in ZA to a minimum of 5.1 pg/m^3^ in SO. In the case study localities in 2021, the highest daily mean PCDD/Fs concentration was recorded in CY at 484.13 fg I-TEQ/m^3^, while the lowest was recorded in KO at 238.1 fg I-TEQ/m^3^. Concentrations of 2,3,7,8-TCDD were highest in CY at 118.53 fg I-TEQ/m^3^, and lowest in KO at 52.38 fg I-TEQ/m^3^. The maximum concentration of dl-PCBs was 48.4 fg I-TEQ/m^3^ in CY and the minimum was 32.4 fg I-TEQ/m^3^ in KO. Similarly, the highest ndl-PCB value was recorded in CY at 6.25 pg/m^3^, and the lowest in GN at 6.10 pg/m^3^.

During the measurements taken in the case study area in 2022/23, the highest concentration of PCDD/Fs was recorded in CY at 203.17 fg I-TEQ/m^3^, while the lowest concentration was found in KZ at 138.1 fg I-TEQ/m^3^. CY also recorded the highest concentration of 2,3,7,8-TCDD at 19.25 fg I-TEQ/m^3^, while GN and KO recorded the lowest concentration at 16.67 fg I-TEQ/m^3^. GN recorded the highest concentration of dl-PCBs at 7.6 fg I-TEQ/m^3^, while KO recorded the lowest at 4.13 fg I-TEQ/m^3^. GN recorded the highest daily average concentration of n-dl-PCBs at 44.74 pg/m^3^, while CY recorded the lowest at 26.26 pg/m^3^.

An assessment of the differences in the concentrations of 2,3,7,8-TCDD and the PCDD/F and dl- and ndl-PCB mixtures between the study villages revealed no statistically significant variation in the levels of air pollution from any of the compounds studied in the individual villages (Table 1).

### 3.2. Population Mortality in the Case Study Area

Between 2005 and 2022, the mean MR of inhabitants in the villages of the case study area was 12.1 per 1000. KO had the lowest rate (MR) at 9.9 per 1000 inhabitants, while Gniazdów had the highest rate at 15.2 per 1000 inhabitants. Among the municipalities, Kochanowice had the lowest MR at 9.8/1000 inhabitants, while Koziegłowy had the highest at 13.1/1000 inhabitants. The MR among females was 10.8 per 1000 women. The lowest rate was recorded in KO at 8.3 per 1000 women, while the highest was in GN at 14.2 per 1000 women. At the municipal level, the MR was lower in KC (8.9 per 1000 women) and higher in KZ (11.7 per 1000 women). The average MR among males was 13.5 per 1000 inhabitants, with the lowest rate observed in KO (11.7 per 1000 males) and the highest in GN (16.2 per 1000 males). Similar to females, male MRs were lower in KC (10.6/1000) and higher in KZ (14.5/1000). The calculations (Table 2) did not demonstrate the statistical significance of the correlation between the MR and the time of the study between 2005 and 2022.

Based on the results of the correlation analysis, it can be concluded that there was no significant change in MRs during the observation period. Therefore, the time factor was excluded from further statistical analysis. As part of this analysis, the study compared MRs between villages and municipalities. The normality of the MR distribution was analysed using the Shapiro–Wilk test, regardless of sex and locality. The results of this test showed that there was no reason to reject the null hypothesis of normality of distribution in each case. Therefore, parametric tests such as the one-way ANOVA test of homogeneity of means could be applied. Based on the results of the post hoc tests, it is evident that the MR of women in the KO village is significantly lower than in the GN village. Similarly, the results of both post hoc tests in the male population of the surveyed villages agree, indicating that the male MR in the GN village is significantly higher than in the KO village. Appendix A show that the MRs of females and males in the study villages were analysed using the Brown–Forsythe test, as well as the Brown–Forsythe and Games–Howell post hoc tests.

Figure 2 shows the mean reaction times (MRs) of the inhabitants of the studied villages. The classic one-factor ANOVA variant indicates significantly different MRs (homogeneity of variance is shown by the Levene test). The Sheffe’s post hoc test reveals that the MR of GN residents is unambiguously higher than that of KO villagers. According to the Fisher LSD post hoc test, the MR in GN is significantly higher than in the other two hamlets. The statistical analysis compared MRs for women, men and all residents in the municipalities of KZ and KC. The results are presented in Figure 3. The non-parametric Mann–Whitney test was used as normality of the MR distribution could not be verified in all cases. The results of this test are clear: the MR in the KZ municipality is significantly higher than in the KC municipality. This conclusion applies to both genders and the entire population. Figure 3 presents mean values and standard deviations.

### 3.3. The Life Expectancy of the Population in the Case Study Area and Silesia Province

In the Silesia Province, the average MLE at birth for the general population in cities with district rights was 71.4 years. The lowest MLE was found in ZO (69.2 years), while the highest was found in BB (74.1 years). The MLE for men was 67.9 years, while for women it was 74.9 years. The shortest MLE for men was found in SW (65.9 years), while the longest was found in BB (70.6 years). Conversely, the lowest MLE among women was in JZ at 72.0 years, while the highest, as with men, was in BB at 77.7 years. The study revealed that the actual MLE of inhabitants in the localities comprising the case study area averaged 74.5 years between 2005 and 2022, which was higher than in cities with county rights. Kochcice had the lowest actual MLE at 71.8 years, while Gniazdów had the highest at 77.5 years. When analysing MLE by gender, it was discovered that the female population had an actual MLE of 79.4 years. The lowest MLE was found in KO at 77.2 years, while the highest was found in GN at 81.6 years. Among men, the average actual MLE was 69.6 years. The lowest MLE was observed in KO (66.3 years), while the highest was observed in GN (73.4 years). A correlation analysis was conducted between the actual MLE and the 18-year time period (2005–2022). Only one statistically significant correlation was found (see Table 3): between the actual MLE of women in the municipality of KC and the time period studied. It can therefore be concluded that the actual MLE did not increase over time in the localities studied during the analysed period.

Based on the results of the correlation analysis, it was assumed that the actual MLE did not change significantly during the observation period. This assumption was used to justify excluding the time factor from further statistical analysis. Appendix A show a comparison of the actual MLE for women (Appendix A) and men (Appendix A) in the study villages.

The Kruskal–Wallis test results (Table 4) clearly indicate statistically significant differences in actual MLE for both men and women. Women with GN have a significantly longer life expectancy than those with KO. This is evidenced by concordant post hoc test results. Additionally, the Dunn–Bonferroni test results indicate that women from GN have a longer life expectancy than women from CY. In contrast, men from GN have a significantly longer MLE than men from KO. The results of both post hoc tests are consistent in this respect.

Table 5 shows the results of a two-means *t*-test comparing MLE between men and women in the municipalities of KZ and KC. The test indicates that MLE is higher in the KZ municipality than in the KC municipality for both men and women.

### 3.4. Impact of Long-Term Inhalation Exposure of the Population to PCDD/Fs and PCBs on Mortality

Due to data access limitations, only socio-communities and communes were analysed to assess the impact of dioxin, furan and polychlorinated biphenyls concentrations on mortality. Actual MLE calculations also took into account 19 cities in the Silesia Province with district rights. The analysis used multiple regression. Preliminary calculations of correlation coefficients between the independent variables were conducted to exclude pairs of strongly correlated independent variables from being present simultaneously in a multiple regression equation. The results of this preliminary correlation analysis are summarised in Table 6. The strongest pairwise correlations are related to the PCDD/Fs and dl-PCBs variables and their respective components, as shown in Table 6. However, the multiple regression analysis excluded these variables.

It was assumed that the multiple regression equation had to satisfy two conditions: the coefficient of determination (R^2^) had to be significantly greater than zero, and each coefficient in the equation had to be significantly different from zero. A significance level of *p* = 0.05 is typically employed in the calculations. MRs were calculated separately for women, men and the general population. The results showed that MR depended significantly only on exposure to 2,3,7.8-TCDD (as shown in Figure 4). The regression equation describes the relationship between MR and 2,3,7.8-TCDD concentrations for women, men and the general population. The corresponding correlation is statistically significant, indicating that the regression line has a slope that is significantly greater than zero.

### 3.5. The Impact of LONG-Term Inhalation Exposure of the Population to PCDD/Fs and PCBs on Life Expectancy

A multiple regression analysis was conducted to determine actual MLE. The resulting equation is as follows:$$MLE=0.04\cdot \frac{PCDD}{Fs}-0.173\cdot 2.3.7.8TCDD+7.84\cdot Sex+69.$$

And all coefficients in the equation yielded *p* < 0.001. The coefficient of determination, R^2^ = 0.641 (*p* < 0.001). For calculation purposes, gender was coded, with a value of 1 for women and 0 for men. The MLE for women was found to be 7.84 years (7 years and 10 months) greater than for men. It is important to note the harmful effect of the 2,3,7.8-TCDD congener. An increase in the concentration of this compound by every 10 fg I-TEQ/m^3^ potentially results in a decrease in MLE, excluding external causes, of 1.7 years (1 year and 9 months). These results could be an effect of inhalation exposure to air pollution with other different factors, like demographic or socioeconomic variables.

## 4. Discussion

The detrimental impact of air pollution on public health is a well-known and extensively documented issue globally, as indicated by increased morbidity rates of numerous diseases and premature mortality in society. To date, scientific attention has focused primarily on pollutants such as particulate matter, nitrogen oxides, sulphur dioxide, benzo[a]pyrene and tropospheric ozone, among others [32,33]. However, the role of inhaling halogenated persistent organic pollutants, such as dioxins, furans and polychlorinated biphenyls (PCBs), in causing various diseases has been explored less thoroughly [34]. Of the 75 polychlorinated dibenzo-p-dioxin congeners and 135 polychlorinated dibenzofuran congeners, 17 compounds are of particular concern due to their toxicity. The most hazardous of these are 2,3,7,8-tetrachlorodibenzo-p-dioxin (TCDD) and 1,2,3,7,8-tetrachlorodibenzo-p-dioxin (PeCDD), which both have a toxicity equivalency factor (TEF) of 1.0, indicating that they are the most toxic compounds in this group of pollutants [35,36,37,38].

As the results of this study indicate, the village of Cynków and its surrounding areas are an example of a region where elevated concentrations of PCDD/Fs and PCBs are potentially emitted from households into the atmosphere. The ongoing practice of local residents disposing of post-production waste containing PVC may result in particularly high exposure to POPs by inhalation during the heating season. According to studies conducted in 2021 in both the case study area and 19 cities with county rights in the Silesia Province, the PCDD/F concentration in Cynków (village) was comparable to that recorded in Zabrze (city). It should be noted that Zabrze has numerous industrial sources of emissions, including a coking plant, a combined heat and power plant, and metallurgical plants, as well as heavy automobile traffic. In contrast, Cynków is a small village surrounded by fields with only one access road and no industrial facilities. For comparison, it is also worth mentioning that, of the aforementioned Silesian cities, Sosnowiec had the lowest concentration of PCDD/Fs, which was more than 2.5 times lower than in Cynków. Within the group of localities forming the case study area, Kochcice had the lowest air pollution from the studied compounds. Subsequent studies conducted at the turn of 2022/2023 also showed the highest concentrations of PCDD/Fs in Cynków and the lowest in Koziegłowy, a municipal town. Interestingly, Cynków is located just a few kilometres from Koziegłowy, yet the indicated differences in halogenated POPs concentrations reached nearly 50%. The results obtained in the case study area showed significantly higher concentrations of PCDD/Fs in the villages of CY and GN than in the reference area (the municipality of KC).

The first air quality studies in the village of Cynków and selected cities in the Silesia province were conducted a decade ago. At that time, it was found that, during the heating season, the concentration of dioxins and furans in the air in Cynków was twice that in Zabrze, the most polluted city in Silesia [9]. However, a different conclusion was reached in a 2019 study by Lopez et al. [39], which was conducted in industrial, urban and reference areas in the Valencia region of Spain. The control area, which lacked significant sources of halogenated POP emissions, exhibited an air pollution level of PCDD/Fs and dl-PCBs of about 3 fg I-TEQ/m^3^, while the concentration of these compounds in other measurement points reached a maximum of more than 300 fg I-TEQ/m^3^. Therefore, the air pollution in Cynków is comparable to the levels of dioxins, furans, and dl-PCBs observed in urban and industrial areas of Spain. This highlights the distinctive nature of the area being studied.

As indicated by the results of air pollution measurements conducted in South Korea from 2008 to 2017, the concentrations of PCDD/Fs and dl-PCBs were significantly lower in urban centres compared to the case study area of this study. Long-term air monitoring in South Korea has revealed a substantial decrease in halogenated POP air pollution. For instance, between 2009 and 2017, the concentration of PCDD/Fs in the atmosphere decreased by an average of 77% (from 51.0 to 14.0 fg I-TEQ/m^3^). This improvement in air quality is the result of the Ministry of Environment’s consistent implementation of the provisions of the Stockholm Convention. Furthermore, nationwide monitoring of 17 PCDD/F congeners and 12 dl-PCB congeners introduced in South Korea in 2008 effectively reduced concentrations of these compounds by over 70% compared to 2001 levels. Therefore, South Korea’s approach could be considered a best practice example for countries such as Poland to follow.

Sources of literature indicate a causal relationship between long-term exposure to dioxins, furans and dl-PCBs through inhalation and increased morbidity and mortality [13,40,41]. Furthermore, in 2012, the International Agency for Research on Cancer (IARC) identified a link between human exposure to TCDD and malignant tumours [42].

The results of this study showed that the MRs for women, men and the overall population differ significantly between the villages of CY and GN and the reference village (KO). Significant differences were also observed between the populations of the municipality under study (KZ) and the control municipality (KC). The MR for the overall population and for each gender in the studied villages (CY and GN) was significantly higher than in the reference village (KO) and the control municipality (KC). The MR of the studied population was 50% higher than that of the reference population. The statistically significant correlations obtained in the study clearly indicate that, as the concentration of 2,3,7.8-TCDD in atmospheric air increases, so do the MRs for women, men and the overall MR. The regression lines depicting the relationship between the MR and the concentration of 2,3,7.8-TCDD for women, men and both genders have a significantly positive slope. Xu et al. [43] reached similar conclusions in a meta-analysis of the impact of inhalation exposure to 2,3,7,8-TCDD on cancer incidence and mortality. Their research demonstrated a statistically significant link between respiratory exposure to 2,3,7,8-TCDD, its concentration in the blood, and mortality caused by overall malignant cancer incidence. An interesting 10-year retrospective study was conducted by Ji et al. [44] in Jiangsu Province in eastern China. Their study aimed to assess the effects of exposure to PCDD/Fs on the population living within a 3 km radius of a municipal solid waste incinerator (MSWI), depending on the wind direction. They found that residents of areas downwind of the incinerator had higher age-adjusted MRs and higher concentrations of PCDD/Fs in their blood serum than residents of areas located on the opposite side of the incinerator.

Another effect of inhaling halogenated POPs, analysed in this study, was a longer mean life expectancy (MLE) among residents of the studied areas. The results suggest that, paradoxically, both women and men from areas with higher levels of PCDD/Fs and PCBs contamination (GN and CY) had a longer MLE than women and men from the less contaminated reference population (KO). However, residents of the entire municipality of KC, where stronger air pollution was recorded, had a shorter MLE than residents of the entire municipality of KZ, where the concentrations of the studied compounds were lower. Exposure to PCDD/Fs and PCBs among residents of the village of KO and the entire reference municipality of KC was similar during the study period. In the municipality of KZ, however, there was significant variation in exposure to PCDD/Fs and PCBs between residents of the villages CY and GN, and those in the rest of the KZ municipality. These results contradict earlier observations made in this area from 2005 to 2022. The reason for these differences may lie in the complexity of the factors influencing the MLE of the studied population [45].

To illustrate the relationship between inhalation exposure to PCDD/Fs and dl-PCBs and the MLE of the population more clearly, the statistical analyses included residents of 19 cities with county rights in the Silesia Province, as well as residents of the case study and reference areas. Including the urban population significantly increased the size of the data pool, producing more representative results. Calculations demonstrated a statistically significant relationship between the concentration of 2,3,7,8-TCDD in the air and the actual MLE of Silesian residents. An increase in the concentration of this compound by 10 fg I-TEQ/m^3^ due to long-term exposure could result in a reduction in the population’s overall MLE of one year and nine months. While the results did not indicate a decrease in the MLE of residents in the case study area, they suggest that the presence of 2,3,7.8-TCDD in the air significantly shortens the population’s actual MLE. This finding suggests that the community may have experienced significant effects of 2,3,7.8-TCDD in the past, as described in earlier publications about Cynków. The study by Dziubanek et al. [46] demonstrated a significant impact of air pollution from particulate matter (PM10), benzo[a]pyrene and heavy metals on the actual MLE of 3.5 million Silesian residents. The study also found that reducing the average annual concentration of PM10 by just 1 µg/m^3^ increases the average MLE of the overall population by 0.1 years. Another study by Dziubanek et al. demonstrated the significant impact of long-term exposure to PCDD/Fs, dl-PCBs, and PM10 on the MLE of the population in 12 cities with county rights in Silesia. The study also showed that reducing the concentration of PCDD/Fs and dl-PCBs by just 10 fg I-TEQ/m^3^ would extend women’s MLE by four months. In a study by Qi et al. [47], the health benefits to the population of implementing measures to achieve the maximum allowable daily concentrations of PM2.5 recommended by the WHO were assessed. The indicator considered was the change in MLE among residents of 72 Chinese cities between 2013 and 2016. The study found that an increase in PM2.5 concentration by 10 µg/m^3^ resulted in an increase in lost life years of over 5 months (0.43 years). Literature data also indicate a significant relationship between society’s exposure to PCDD/Fs and PCBs through inhalation and cancer morbidity and mortality. For example, Zheng et al. [48] conducted a study on a population of 2992 adult residents of China from 1999 to 2004. The results showed that a 1-logarithmic-unit increase in the concentration of 1,2,3,4,6,7,8,9-octachlorodibenzo-p-dioxin (1,2,3,4,6,7,8,9-OCDD) resulted in a 76% increase in the probability of death from any cause. Meanwhile, a 1-logarithmic-unit increase in the concentration of 1,2,3,4,6,7,8-heptachlorodibenzofuran (1,2,3,4,6,7,8-HpCDF) could potentially critically increase the risk of death from malignant cancer by up to 90%. These results indicate that improving air quality translates into tangible health benefits for society. This information is particularly important for local and central governments, as they should be aware that funds allocated to improving air quality will lead to better public health and reduced healthcare costs in the future [49,50].

The study has its limitations. Firstly, this manuscript presents the results of an ecological study relating community-level exposure to mortality indicators. It is well known that, among observational studies, environmental studies represent the lowest level of evidence, as they are prone to ecological bias. This means that the level of association cannot be translated into an individual-level association due to an imbalance of confounders among the study units. Nevertheless, this type of research enables the impact of environmental factors on society’s health and quality of life to be assessed and also allows hypotheses for further analytical research to be generated. It should be emphasised that the ecological correlations obtained in this study cannot confirm relationships at the individual level directly, but can indicate the need for further, more detailed research. Furthermore, there is a potential possibility of selection bias in the data obtained from national statistical offices.

In addition, the difficulty of this research may have resulted in a limited number of air samples being collected for analysis. Due to budget constraints and the high cost of laboratory analyses related to determining individual POPs in air samples, the authors were forced to limit the number of samples analysed. However, they decided to double the number of samples in the study and control areas in order to thoroughly analyse local air pollution with POPs. Nevertheless, the study area was relatively small, and analysis of the results obtained from such a small number of environmental samples indicates significant air pollution with POP compounds in the study area. Increasing the number of air samples would confirm the conclusions obtained in this study.

Another limitation of this study was the small size of the study and control areas (villages CY, GN and KO), which have small populations. Consequently, the number of deaths without external causes that we analysed was limited to a few cases per year across the entire study population. This prevented us from analysing the number of deaths by age group, which would have added value to this study. Furthermore, statistical analysis of the relationship between mortality and air pollution is extremely difficult in such cases. Therefore, these studies should be carried out over a larger area in the future, with an appropriate budget, and should cover areas characterised by significant POPs air pollution.

A subsequent limitation is the lack of consideration of the potential impact of socio-economic factors, access to healthcare and lifestyles in the statistical analysis. The geographical proximity of the case study and reference areas, coupled with the prevalence of strong family ties among their inhabitants, has played a critical role in shaping these communities’ comparable characteristics. Regarding the 19 cities with county rights in the Silesia Province, it is evident that the socioeconomic conditions are less homogeneous than in the villages studied. However, due to their close geographical proximity, these cities often border each other directly, enabling residents to move between urban areas via the dense road network and well-developed public transport system. Evidence of this resident mobility is apparent in various aspects of daily life, including employment, education (schools and universities), healthcare (e.g., highly specialised regional hospitals) and recreational activities. Consequently, it can be concluded that the socioeconomic conditions, lifestyle factors, and healthcare services under discussion are comparable in this case.

This study builds upon previous research in this area. Dziubanek et al.’s (2016) preceding study was conducted in just five cities and estimated the inhalation exposure of adults and children to PCDD/Fs and dl-PCBs, as well as the associated cancer and non-cancer health risks [12]. Subsequent research by Dziubanek et al. (2017) considered the combined impact of PCDD/Fs, dl-PCBs and PM10 on the MLE of residents in Cynków and 12 other towns [13]. However, the present study encompasses not only the designated case study area, but also the reference area. Furthermore, it involved 19 cities in the Silesian Voivodeship, and statistical analyses examined residents’ exposure to a mixture of halogenated persistent organic pollutants (POPs), as well as to the specific congener 2,3,7,8-tetrachlorodibenzo-p-dioxin (TCDD). The present study goes beyond analysing MLE; it also analyses mortality within the population. Consequently, this research should be regarded as innovative, substantially expanding existing research.

The objective of the present study was to examine the impact of inhalation exposure on MLE and MRs. However, it is acknowledged that exposure levels to dioxins, furans and PCBs are also influenced by contamination of the environment (e.g., food, water and soil) with these compounds present. Nevertheless, due to financial constraints, an integrated assessment of the population’s exposure to halogenated POPs, taking into account their presence in local organisms (e.g., adipose tissue), was not possible to implement [51]. The high cost of laboratory tests led to the decision to assess the impact of inhalation exposure, which is a significant consideration in the context of the selected case study area. Nevertheless, a few years ago, the authors conducted studies analysing the levels of PCDDs, PCDFs and PCBs in free-range chicken meat and eggs, as well as cow’s milk, produced on farms in Cynków. Control samples were obtained from local grocery stores and consisted of the same foods. The analysis demonstrated that the compounds in food products from the case study area exhibited much higher concentrations than those found in food products from stores. Exposure to concentrations of polychlorinated dibenzo-p-dioxins (PCDDs), polychlorinated dibenzofurans (PCDFs) and dioxin-like polychlorinated biphenyls (dl-PCBs) has been shown to increase the risk of cancer and non-cancer health issues. The hazard quotient (HQ) was found to be 71.3 for non-cancer risk and 7.5 × 10^−3^ for cancer risk. The health risks posed by exposure to these compounds in food items purchased from stores in the study area were found to be lower, with an HQ of 0.8 and a cancer risk of 6.9 × 10^−4^ [52]. No calculations were made in the present study pertaining to the cause-and-effect relationship between population exposure through the oral route and MLE and MRs. This is because data on food contamination in the village of Cynków is unavailable. It is therefore recommended that future studies expand on the current research by incorporating additional exposure factors such as food, water and soil contamination, as well as socioeconomic factors and the lifestyles of inhabitants. Local monitoring of POP emissions into the environment is also necessary, as is raising public awareness in this regard. Additionally, analysing the presence of halogenated POPs in inhabitants’ tissues should be considered. This approach will facilitate a comprehensive analysis of the combined impact of these compounds on MRs and MLE.

## 5. Conclusions

The ecological research revealed a significant correlation between the chronic inhalation exposure of residents of the case study area to 2,3,7.8-TCDD and their overall mortality, excluding external causes of death. This very strong correlation indicates the markedly negative impact of 2,3,7,8-TCDD on health, affecting women, men and the overall population. Furthermore, including inhalation exposure to a mixture of PCDDs/PCDFs and dl-PCBs in a population sample from 19 cities with county rights in Silesia demonstrated a strong correlation between the concentration of 2,3,7,8-TCDD in the air and the actual mean life expectancy (MLE) of Silesia residents. Due to long-term exposure to high levels of POPs, an increase in the concentration of the 2,3,7,8-TCDD congener by 10 fg I-TEQ/m^3^ could result in a reduction in the MLE of exposed residents of up to one year and nine months. Long-term exposure to dioxins, furans and polychlorinated biphenyls (PCBs) may also result in a significant disparity in the MLE of women and men in Silesian areas. On average, women in the studied areas live 7 years and 10 months longer than men. However, considering the considerable differences in mean MLE observed even in the low-polluted areas, it is likely that not only environmental factors, including exposure to PCDD/Fs and PCBs, but also various socio-economic factors may be involved.

## Figures and Tables

**Figure 1 toxics-13-01002-f001:**
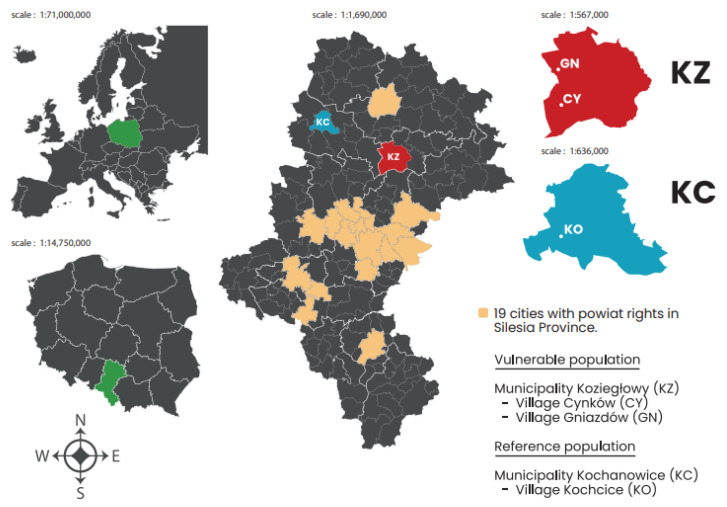
Poland and Silesia with marked municipalities and villages.

**Figure 2 toxics-13-01002-f002:**
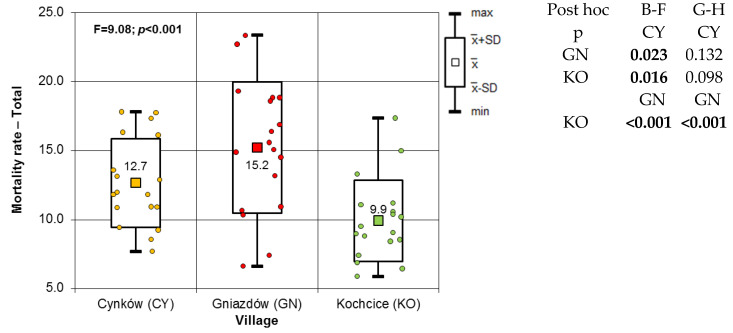
The MRs of the population in the study villages.

**Figure 3 toxics-13-01002-f003:**
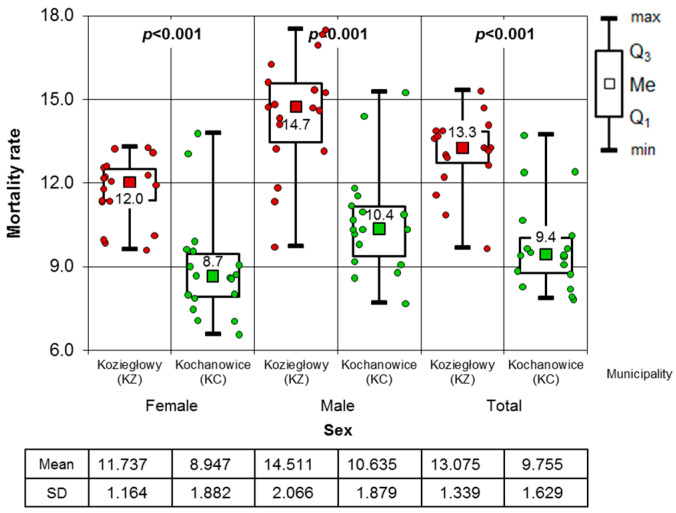
The MRs for females, males and the total population in the municipalities of Koziegłowy and Kochanowice.

**Figure 4 toxics-13-01002-f004:**
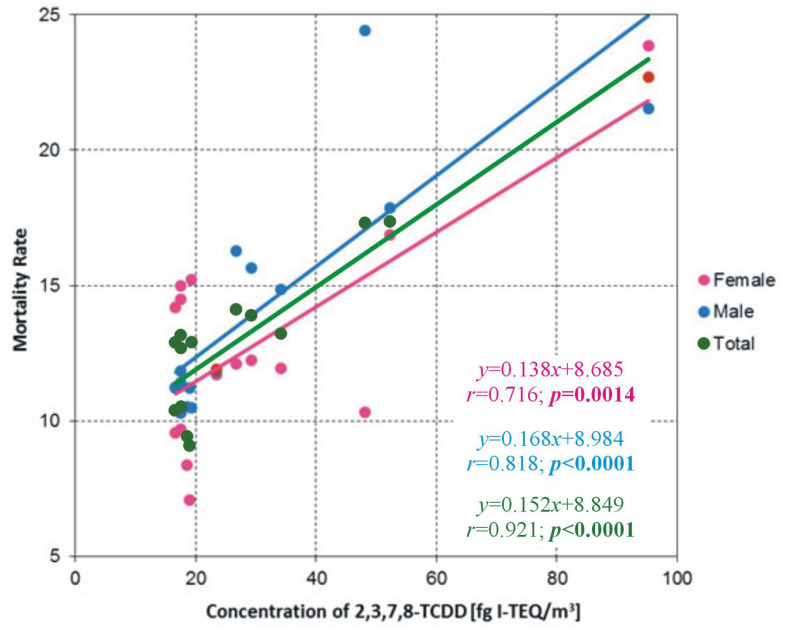
The relationship between MR and 2,3,7.8-TCDD concentrations for women, men, and the total population.

**Table 1 toxics-13-01002-t001:** Assessment of the significance of differences in concentrations of dioxins, furans and polychlorinated biphenyls between the studied villages—results of the Kruskal–Wallis test.

Location	PCDD/Fs	dl-PCBs	PCDD/Fs and dl-PCBs	ndl-PCBs	2,3,7.8-TCDD
	[fg I-TEQ/m^3^]	[fg I-TEQ/m^3^]	[fg I-TEQ/m^3^]	[pg/m^3^]	[fg I-TEQ/m^3^]
Heating season: December–February 2014/2015					
CY	185.60	12.8	198.40	32.00	41.20
KZ	110.90	9.40	120.30	27.90	29.23
Heating season: February–March 2021					
CY	484.13	48.40	532.53	6.25	118.53
GN	357.14	43.50	400.64	6.10	95.24
KO	238.10	32.40	270.50	6.19	52.38
Heating season: December–February 2022/2023					
CY	188.89	4.19	193.08	26.26	19.25
CY	203.17	5.10	208.28	33.71	17.52
GN	182.54	7.60	190.14	41.00	17.46
GN	176.19	5.56	181.75	44.74	16.67
KO	178.57	6.37	184.94	34.16	17.46
KO	169.05	4.13	173.18	30.83	16.67
KZ	138.10	4.33	142.43	38.95	26.78
KC	178.57	6.02	184.60	36.99	19.05
Results of the Kruskal–Wallis test (comparison between villages)					
Statistics Kruskal–Wallis	1.800	0.345	1.800	0.636	0.755
*p*-value	0.433	0.869	0.433	0.791	0.715

**Table 2 toxics-13-01002-t002:** Assessment of the statistical significance of the correlation between MRs and the timing of the surveys (2005–2022).

Location	Female		Male		Total	
	r	*p*	r	*p*	r	*p*
CY	0.017	0.947	–0.440	0.068	–0.354	0.15
GN	0.130	0.606	–0.199	0.428	–0.040	0.874
KO	0.358	0.145	0.352	0.152	0.447	0.063
KZ	0.144	0.570	–0.464	0.052	–0.286	0.249
KC	0.183	0.468	0.448	0.062	0.379	0.121

**Table 3 toxics-13-01002-t003:** Assessment of the statistical significance of the correlation between MLE (in years) and follow-up time from 2005 to 2022.

Location	Female		Male	
	r	*p*	r	*p*
CY	–0.009	0.970	0.235	0.348
GN	0.042	0.867	0.134	0.596
KO	0.183	0.467	0.269	0.281
KZ	0.385	0.114	0.421	0.082
KC	0.516	0.028	0.401	0.099

**Table 4 toxics-13-01002-t004:** The life expectancy of males and females in the analysed villages.

Sex	Village	Mean	SD	Min	Q1	Median	Q3	Max	KW Test
Male	CY	70.6	5.82	56.8	68.5	69.9	71.2	83.9	KW = 10.55 *p* = 0.0051
	GN	73.4	5.20	66.9	68.6	72.9	77.9	81.8	
	KO	66.3	5.70	51.5	63.0	66.2	69.8	75.4	
Female	CY	80.0	5.34	64.3	78.2	80.8	82.7	88.2	KW = 13.98 *p* = 0.0009
	GN	81.6	6.87	58.0	79.5	83.0	85.4	89.0	
	KO	77.2	3.08	69.5	75.7	77.1	79.0	84.0	

**Table 5 toxics-13-01002-t005:** The life expectancy of males and females in the analysed municipalities.

Sex	Municipality	Mean	SD	Min	Q1	Median	Q3	Max	Two Sample *t* Test
Male	KZ	69.9	1.42	66.9	69.3	69.8	70.8	72.3	*p* = 0.002
	KC	67.8	2.47	62.9	66.5	67.8	69.3	72.3	
Female	KZ	80.0	1.22	77.2	79.3	79.9	80.6	82.3	*p* = 0.012
	KC	78.2	2.79	70.2	77.4	78.8	79.5	82.4	

**Table 6 toxics-13-01002-t006:** Cross-correlation coefficients between independent variables.

	dl-PCBs	PCDD/F and dl-PCBs	n-dl-PCB	2,3,7.8-TCDD
PCDD/Fs	0.888	0.998	–0.712	0.317
dl-PCBs		0.915	–0.893	0.608
PCDD/Fs and dl-PCBs			–0.746	0.360
ndl-PCBs				−0.619

## Data Availability

The data presented in this study are available upon request from the corresponding author. The data are not publicly available due to privacy restrictions.

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
