# Peer review of "Life Expectancy and Mortality in the Aspect of Diverse Environmental Exposure to PCDD/Fs and PCBs—Ecological Case Study from the Silesia Province, Poland"

_toxics, 2025, doi:10.3390/toxics13111002_

Round 1

Reviewer 1 Report (Previous Reviewer 3)

Comments and Suggestions for Authors

I think he authors made a good effort on replying to my previous comments.

I think the study has still some limitations, but those are now recognised in the discussion.

I have a doubt on the regression presented in lines 441-460. The authors presented the results for the regression with mortality rate as outcome Line 443, but the equation in line 453 is for life expsctancy as outcome.

Author Response

Dear Reviewer,

We would like to thank you very much for accepting the corrections made, in accordance with the Reviewer's previous comments.

In response to the additional comment of the Reviewer, which reads: "I have a doubt on the regression presented in lines 441-460. The authors presented the results for the regression with mortality rate as outcome Line 443, but the equation in line 453 is for life expectancy as outcome.", we would like to inform you that we have separated the results regarding mortality from the results regarding life expectancy by introducing an additional subsection 3.5 (lines: 477-478). We thank you for the above suggestion and hope that our correction will allow the reader to understand the results more precisely.

Sincerely,

Authors

Reviewer 2 Report (New Reviewer)

Comments and Suggestions for Authors

The manuscript presents an important and original study addressing the impact of environmental exposure to polychlorinated dibenzo-p-dioxins (PCDD/Fs) and polychlorinated biphenyls (PCBs) on life expectancy and mortality rates in the Silesia Province of Poland. The authors have conducted a case study involving small neighboring villages with distinct exposure profiles and supplemented this with data from 19 cities in the region. The study is relevant to public health and environmental toxicology and has the potential to contribute meaningfully to understanding localized effects of persistent organic pollutants (POPs) on population health.

Overall, the topic is valuable and timely, and the manuscript is well organized, but there are areas where clarity, scientific rigor, and language precision could be improved. The study’s ecological design and limited sample represent a valid exploratory approach; however, the interpretation of causal relationships requires caution. The manuscript could benefit from revision for structure, methodological transparency, and conciseness.

Major Comments

1. Novelty and Relevance

  1. The local-scale approach focusing on chronic POP exposure due to domestic waste burning is an innovative aspect. However, the novelty should be highlighted more clearly in the Introduction and Discussion compared with previous works from the same group and region.
  2. Please clarify how this study extends prior findings (e.g., Dziubanek et al., 2015, 2020).

2. Study Design and Methodology

  1. The study uses ecological data, yet at several points the text implies causal inference. The authors should consistently state that ecological correlations cannot confirm individual-level associations.
  2. Sampling frequency and representativeness are limited. Please discuss how seasonal variability and sample size might affect conclusions.
  3. Statistical analysis is comprehensive, but the description is overly technical and repetitive. Consider presenting only essential details in the main text, with full formulas or codes moved to Supplementary Materials.

3. Data Interpretation

  1. The claim that “an increase of 10 fg I-TEQ/m³ of 2,3,7,8-TCDD reduces life expectancy by 1.7 years” should be qualified carefully—this relationship is statistical, not mechanistic. The Discussion should emphasize that confounding factors (e.g., socioeconomic status, healthcare access, comorbidities) were not controlled for.
  2. The paradoxical finding of longer life expectancy in more contaminated villages (GN and CY) than in the reference area (KO) should be more critically interpreted. Possible confounders or sampling artifacts must be discussed more explicitly.

4. Figures and Tables

  1. Figures 2–9 are informative but overly complex. Consider combining or simplifying some figures and improving the readability of axis labels and legends.
  2. Include units and confidence intervals consistently across all tables.

5. Language and Clarity

  1. The manuscript would benefit from language editing by a fluent English speaker or a professional editing service. Issues include long sentences, redundant phrases, and inconsistent tense.
  2. Example: Replace “the problem is more crucial” with “the problem is more critical” or “the issue is more pronounced.”
  3. Ensure consistent use of abbreviations (e.g., PCDD/Fs, POPs, MR, MLE) and define them upon first use.

6. Discussion and Context

  1. The Discussion appropriately references relevant literature but could be condensed and more focused on interpretation rather than description.
  2. Highlight the policy implications and potential for intervention—e.g., need for local emission monitoring and community awareness.

7. Limitations

The limitations section is well stated but should include acknowledgment of potential selection bias in the data obtained from national statistical offices and the lack of individual-level exposure verification.

Minor Comments

  1. Abstract: Simplify and clarify sentences for non-specialist readers. Include the sample size and main findings concisely.
  2. Keywords: Add “environmental epidemiology” and “public health risk assessment” for better indexing.
  3. Introduction: Provide a clearer rationale linking POPs exposure and population health metrics.
  4. References: Verify formatting consistency with Toxics journal style (DOI, capitalization, and order).
  5. Units: Use consistent presentation of units (e.g., fg I-TEQ/m³) throughout the text.
  6. Grammar examples:
  1. “air pollution causes many environmental health problems in the world, regardless income status” → “Air pollution causes numerous health problems globally, regardless of income level.”
  2. “where coal and non-ferrous metal ore mines have been operating for many years” → “which has long been an area of coal and non-ferrous metal mining.”

Author Response

Dear Reviewer,

We would like to sincerely thank you for your detailed review of our publication and for pointing out areas where we could improve it. We have tried to implement all the changes suggested by the reviewer to the best of our ability. We hope you will be satisfied with our improvements. We have attached a file with answers to all your questions and comments.

Sincerely,
Authors

Reviewer 3 Report (New Reviewer)

Comments and Suggestions for Authors

Comments on

Life Expectancy and Mortality in the Aspect of Diverse Envi-2 ronmental Exposure to PCDD/Fs and PCBs - Case Study from 3 the Silesia Province, Poland

Manuscript Number: toxics-3935253

General comments:

This study focuses on the impact of long-term exposure to polychlorinated diphenylene oxides (PCBs) and dioxins (PCDDs/Fs) on life expectancy and mortality rates in the Silesian Province of Poland. The research topic has practical significance in public health and environmental epidemiology, and is particularly valuable for reference in the field of air pollution and health risk assessment in Europe. However, I think the quality of the manuscript needs to be further improved, although the issues that are being addressed may be of interest to our readers.

Specific comments:

  1. The overall structure is reasonable, but there is some redundancy and repetition, especially in the discussion section where results are repeatedly cited. The number of figures and tables is excessive (>10); it is recommended to merge or simplify them.
  2. Similar regional exposure-health association studies have been reported in Eastern Europe and China. The novelty of this paper lies in the micro-scale analysis at the village level, but it lacks a clear exploration of the mechanisms or methodological innovation.
  3. The abstract is too long, and the innovative points are not highlighted. It is recommended to shorten it to within 250 words and clearly state the innovative points.
  4. The regression model only explains the difference in life expectancy based on pollutant concentration, without considering demographic or socioeconomic variables, resulting in weak causal inference. The conclusion that "for every 10 fg I-TEQ/m³ increase, life expectancy decreases by 1.7 years" seems to be an over-extension and should be expressed with caution.
  5. While the small sample size and high cost have been mentioned, the factor of population confounding has not been addressed. The potential impacts of socioeconomic factors, lifestyle, and differences in health services should be supplemented.

Author Response

Dear Reviewer,

We would like to sincerely thank you for your detailed review of our publication and for pointing out areas where we could improve it. We have tried to implement all the changes suggested by the reviewer to the best of our ability. We hope you will be satisfied with our improvements. We have attached a file with answers to all your questions and comments.

Sincerely,
Authors

Round 2

Reviewer 2 Report (New Reviewer)

Comments and Suggestions for Authors

The authors have thoroughly and effectively addressed all reviewer comments from the previous round. The revised version of the manuscript, “Life Expectancy and Mortality in the Aspect of Diverse Environmental Exposure to PCDD/Fs and PCBs – Case Study from the Silesia Province, Poland,” demonstrates substantial improvement in both scientific clarity and presentation quality.

The responses provided are comprehensive, and all methodological, analytical, and interpretative points raised earlier have been incorporated appropriately. The structure is now coherent, the results are clearly presented with proper statistical support, and the discussion effectively integrates the findings with relevant literature. Figures and tables have been refined, and the English language has been noticeably improved, making the manuscript clearer and easier to follow.

The inclusion of detailed descriptions of the study area, data sources, and regression analysis strengthens the scientific rigor and reproducibility of the research. Furthermore, the expanded discussion and clarification of limitations enhance the overall credibility and transparency of the study.

Author Response

Dear Reviewer,

We sincerely thank you for appreciating our efforts to improve the manuscript in accordance with your guidelines. We are confident that the changes have made our results more clearly presented and significantly easier to read. This is all thanks to your suggestions.

Thank you again for your time and valuable comments.

Authors

This manuscript is a resubmission of an earlier submission. The following is a list of the peer review reports and author responses from that submission.

Round 1

Reviewer 1 Report

Comments and Suggestions for Authors

The article is interesting and addresses an issue that is of concern today, given the increase in pollution. However, it needs to be corrected in its presentation, as the tables are cut off and misplaced in the text. 
It is recommended that a section on the limitations of the study and future prospects be included.

Reviewer 2 Report

Comments and Suggestions for Authors

The manuscript focuses on the effects of long-term inhalation exposure to PCDD/Fs and PCBs on longevity and mortality in populations. Although the research has important implications for micro-scale environmental health research in the Silesian region of Poland, some methodological details, interpretation of results and discussion of limitations still need to be strengthened.

1. This article is of limited innovation, and how it differs from similar studies. It is recommended to highlight the unique innovations of this study such as special exposure scenarios of rural PVC waste burning.
2. The study incorporated a relatively small sample size of only one sampling point, which may limit the statistical robustness and wider applicability of its results. In order to increase the credibility of the study's findings, it would be advisable to validate them using a wider sample population. This would greatly improve the reliability of the study's findings and enhance their generalizability.
3. In the introductory section, it is suggested that additional information be provided on the role played by other tools, such as advanced materials in air pollution control, such as Nature Communications, 2019, 10, 1458、ACS Nano, 2019, 13, 11912、Advanced Functional Materials, 2023, 34, 230677.
4. It is recommended that the scale of figure I be supplemented and that the abbreviations be harmonized, such as defining “PCDD/Fs” on first appearance.
5. Some references are full names and some are abbreviations, it is recommended to standardise the format and increase the number of references.

Reviewer 3 Report

Comments and Suggestions for Authors

This paper presents the results of an ecological study evaluating the association between environmental exposure to PCDD/Fs and PCBs with mortality indicators in the Silesia Province, Poland.

The research question is undoubtedly of interest, given the high exposure to the PCCD/Fs and PCBs that the population in the study area is experiencing. Unfortunately, the study design and, more importantly, the statistical methods don’t seem appropriate to address the research question.

  1. In the abstract (line 18), the author stated that they conducted a causal analysis between community-level exposure and mortality indicators. This statement is hard to sustain, as this paper presents the result of an ecological study relating community-level exposure and mortality indicators. It’s well known that environmental studies represent the lowest level of evidence among observational studies, as they are prone to ecological bias, for which the association level cannot translate into individual-level association due to imbalance of confounders among study units. Differences in the mortality indicators could be due to different age distributions and could give a spurious association between spatial units.
  2. The statistical analysis methods lack details and clarity, and they not seem coherent with the data structure:
  3. The analysis of the results by the study area (villages CY, GN and KO, and municipalities KZ and KC) is a bit confusing, given the choice to consider villages and municipalities that could have different levels of measurement error. It is possible that the measurement error would be lower at the village level than the municipality level, given the smaller area to which the measurement refers.
  4. It is unclear how the Kruskal-Wallis test was applied to compare exposure levels between villages (Table 1).
    1. Did the authors consider all the observations for each village? For example, CY has four observations (one in 2014-2016, one in 2021 and two in 2022/2023), GN, and KO have 3 observations (one in 2021 and two in 2022/2023).
    2. Why are there two measurements in 2022/2023 in villages CY, GN and KO?
    3. As a note, I would label the measurement period as February-March 2021, and December 2022-February 2023.
    4. This low number of measurements makes it hard to derive inferences on differences between villages. The statistical test has low power with a high probablity of false report findings.
  5. The analysis of mortality indicators by 3 villages and 2 municipalities has several caveats:
    1. If I understood well, the authors considered crude mortality rates in their analysis, e.g. they didn’t consider age-specific mortality rates. Differences in the age distribution between villages and municipalities could explain the differences observed between villages and municipalities, for example, the highest mortality rate observed in GN could be due to the older population, as described by the higher life expectancy index. The authors should use direct or indirect methods to standardise mortality rates if they want ot compare between locations.
    2. If I understood well, the authors used the mortality rates and life expectancy indexes calculated between 2005 and 2022 and compared them between locations. I have two reservations about this procedure. What is the rationale for including mortality data in years (2021-2023) before the exposure was assessed? The authors didn’t consider possible autocorrelation of the outcome when evaluating possible differences between villages and municipalities. The authors should use methods that consider the longitudinal structure of the data (over years) in their analysis, e.g. repeated measurements ANOVA or fixed effects models.
  6. The analysis of the exposure levels and mortality between 19 cities also has several caveats:
    1. As stated, the 19 cities could differ for several contextual factors explaining the observed association. One is age, which could be easily addressed by calculating standardised mortality rates with direct or indirect methods.
    2. The inferential structure of the regression is based on the normality of the residual assumption. Looking at Figure 9, this assumption is hard to sustain. Most cities seem to have an exposure level around 20, with a few influential points over 20. Is there a detection threshold near 20? The authors should consider transformation of the exposure if they want to use regression, categorise the exposure or use methods that consider detection limits (e.g see Enrique F. Schisterman, Albert Vexler, Brian W. Whitcomb, Aiyi Liu, The Limitations due to Exposure Detection Limits for Regression Models, American Journal of Epidemiology, Volume 163, Issue 4, 15 February 2006, Pages 374–383).
    3. I would change the colours in Figure 9 as it’s hard to distinguish purple from red points.
  7. In the discussion, all the comments on possible differences between villages, municipalities, and cities are hard to sustain, given my comments on study design and statistical methods.

Other minor comments:

In Figure 1, I would indicate the 19 cities included in the analysis. Is it useful to represent the first map of Poland in Central and Eastern Europe? Eventually, I would show all of Europe.